# Biology and Regulation of Staphylococcal Biofilm

**DOI:** 10.3390/ijms24065218

**Published:** 2023-03-09

**Authors:** Patrice François, Jacques Schrenzel, Friedrich Götz

**Affiliations:** 1Genomic Research Laboratory, Service of Infectious Diseases, Geneva and University Hospitals, 1205 Geneva, Switzerland; 2Bacteriology Laboratory, Division of Laboratory Medicine, Department of Diagnostics, Geneva University Hospitals,1205 Geneva, Switzerland; 3Microbial Genetics, Interfaculty Institute of Microbiology and Infection Medicine Tübingen (IMIT), University of Tübingen, 72076 Tübingen, Germany

**Keywords:** staphylococci, biofilm, gene expression, regulation, resistance, tolerance, foreign body, metabolism, phenotype

## Abstract

Despite continuing progress in medical and surgical procedures, staphylococci remain the major Gram-positive bacterial pathogens that cause a wide spectrum of diseases, especially in patients requiring the utilization of indwelling catheters and prosthetic devices implanted temporarily or for prolonged periods of time. Within the genus, if *Staphylococcus aureus* and *S. epidermidis* are prevalent species responsible for infections, several coagulase-negative species which are normal components of our microflora also constitute opportunistic pathogens that are able to infect patients. In such a clinical context, staphylococci producing biofilms show an increased resistance to antimicrobials and host immune defenses. Although the biochemical composition of the biofilm matrix has been extensively studied, the regulation of biofilm formation and the factors contributing to its stability and release are currently still being discovered. This review presents and discusses the composition and some regulation elements of biofilm development and describes its clinical importance. Finally, we summarize the numerous and various recent studies that address attempts to destroy an already-formed biofilm within the clinical context as a potential therapeutic strategy to avoid the removal of infected implant material, a critical event for patient convenience and health care costs.

## 1. Biofilm Formation and Clinical Significance

Most bacterial human diseases involve biofilm-producing pathogens. Bacteria growing in surface-associated communities, which are described as biofilms, are physiologically distinct from free-swimming, planktonic-state organisms. Biofilms can be defined as sessile microbial communities that are embedded in a self-produced extracellular matrix [1,2] of polysaccharidic or proteinaceous nature associated with DNA, yielding to so-called “hydrated surface-associated communities” [3]. While biofilms were first described in aquatic environments, biofilm formation is increasingly recognized as an important parameter in the pathogenesis of many bacterial infections. Among these infections are diseases that involve the formation of a biofilm on the biomaterials frequently used in modern medicine (e.g., catheters and polymeric or metallic implants) and hard mineral surfaces (e.g., teeth and bones) [4,5,6]. The hallmark characteristic of a biofilm is the development of a three-dimensional structure of bacteria that is stabilized within an exopolysaccharide glycocalyx [7]. The formation of bacterial biofilms is an elaborate process composed of four consecutive phases: attachment, accumulation, maturation, and spontaneous dispersal [8]. This complex and structured architecture protects the bacteria from hostile environments such as the human body [9]; this is not the case for free-floating organisms [10]. In addition, a biofilm’s mode of growth provides altered susceptibility to some antimicrobials [11,12,13] and affects bacterial killing by professional phagocytes [14,15]. The 3D structural organization of bacterial biofilms contains bacteria with different phenotypes [16,17] and various growth rates and metabolic activity, yielding a limited efficiency of the antibiotics that target cell-wall biosynthesis while the reduced oxidative metabolism limits the access of aminoglycosides to their target [18].

Biomaterials implanted for a prolonged period of time, such as durable catheters and orthopedic implants, are frequent sources of sepsis and infections, mainly due to slime-producing and biofilm-forming bacteria. Thus, a question arose with respect to the development of materials that demonstrate a reduced incidence of biofilm formation. It was rapidly noticed that bacterial adhesion (attachment) is the first step in biofilm formation, and various methods have been developed to assess adherence and biofilm formation on a given polymer surface. A brief overview was presented in the review by Götz and Peters [19]. Here, the authors showed that coagulase-negative staphylococci and *S. aureus* can bind to almost any implanted material composed of plastic, stainless steel, or titanium (see Table 1). Bacterial adhesion is not dependent on surface type, whether smooth or textured; or on the polymeric composition of the implant material, whether silicone or polyurethane; nor is it dependent on the presence or absence of slime. Bacterial adhesion to biomaterials is a general process that is most likely due to many surface components. With the help of a green fluorescent protein (gfp) reporter plasmid in *S. aureus* [20], adhesion and biofilm formation were investigated on various surfaces (Figure 1). Glass slides were coated with three different materials used as medical devices: titanium, cobalt, and Teflon [21]. As shown, *S. aureus* adhered and formed a biofilm even on titanium, the most frequently used material for hip prosthesis. It also adhered and formed a film on cobalt surfaces, while the adherence to Teflon was less pronounced. 

Biofilm formation appears to be genetically programmed and finely regulated [41,42,43,44], allowing bacteria to control their microenvironment [2,17,45,46] and to actively detach from the biofilm matrix to generate metastatic infectious foci [47]. Genetic analyses were used to reveal the diversity of genetic factors contributing to biofilm formation, and it appears clearly that multiple pathways are involved in building bacterial biofilm [40,48,49,50]. These factors, especially during the early stages of biofilm formation, can be functionally replaced or compensated for by others, depending on environmental and growth conditions [44,46,51,52].

The vast majority of nosocomial infections involved a biofilm-producing organism [2]. Thus, the development of strategies that limit biofilm development by using modified biomaterials or permitting the dissociation of already-formed biofilms in order to avoid material removal constitute urgent clinical needs in situations of economic or clinical relevance [53].

## 2. Molecular Control of *S. aureus* Biofilm Development and the Role of *ica*

The formation of mature, three-dimensional biofilms is a complex process composed of different phases: attachment, accumulation, maturation, and dispersal [54,55]. While the initial binding to abiotic (protein-free) surfaces in vitro is mostly based on hydrophobic interactions, primary attachment during infection occurs via the binding of specific bacterial surface receptors that recognize host matrix proteins [56]. This group of cell-wall-anchored proteins, named MSCRAMMs (for microbial surface components recognizing adhesive matrix molecules) [57,58], presents a conserved structure containing 4–5 domains with the binding domain exposed to the extracellular medium. The accumulation phase appears to be related to the production of polysaccharide adhesins that allow interactions between bacterial cells [42,59]. Thus, the primary determinant of the accumulation phase of staphylococcal biofilm formation relates on the production of the polysaccharide intercellular adhesin (PIA), a process that is dependent on the expression of genes of the *icaADBC* operon [60,61]. Biochemical studies have demonstrated that the PIA consists of polymeric *N*-acetylglucosamine in which the cells are embedded and protected against humoral and cellular host immune defense and against antibiotic treatments [14,62,63]. PIAs act as an intercellular adhesin, allowing for the integration of bacterial DNA [64] and constituting a stable, organized structure. They appear to play a role in the formation of multiple bacterial clusters that are involved in biofilm maturation and include the accumulation-associated protein [42,65] and other proteins, such as clumping factor A (ClfA) [42], the staphylococcal surface protein (SSP1), and the biofilm-associated protein (Bap) [42].

With the increasing number of sequenced genomes due to progress in high-throughput sequencing capacity, an *ica* locus has been identified in several staphylococci species: *S. caprae*, *S. roterodami*, *S. carnosus*, *S. saprophyticus*, *S. cohnii*, *S. capitis*, *S. sciuri*, *S. hominis*, and *S. simulans.* It appears to serve the same function as in *S. aureus*. Note that if *S. aureus* remains a potent human pathogen, most of these species represent potential human opportunistic pathogens.

The *ica* operon was first identified in *S. epidermidis* [39,40] and has been studied most extensively in that species. The *ica* operon is subject to environmental regulation [66]. For example, anaerobic growth was found to induce expression of the *ica* operon and PIA production in both *S. epidermidis* and *S. aureus* [46]. Expression of the *icaADBC* operon appears tightly controlled in *S. aureus*, evidenced by the fact that it is expressed at very low levels under in vitro growth conditions [67]. Beenken et al. found that the mutation of *ica* and the resultant inability to produce PIA had little impact on in vitro biofilm formation or the colonization of an abiotic surface [44]. Our group also compared an *S. aureus* strain and its corresponding *ica* mutant in a tissue cage model of infection and demonstrated that the *ica* mutant retained the capacity to colonize at a similar level to the wild-type strain [68], a result which was confirmed by others [69]. Taken together, the expression of *ica* plays a major role in biofilm formation but is not essential in the colonization of a surface. Interestingly Rachid et al. [70] showed that the expression of *ica* is at least partially controlled by the stress response transcription factor, σ^B^ [71]. Studies performed with *S. aureus* have demonstrated that the regulation of *ica* expression and the ability to form a biofilm involve regulatory elements other than σ^B^ and IcaR [72]. Among these additional regulatory loci, the accessory gene regulator (*agr*) and the staphylococcal accessory regulator (*sarA*) represent important partners. Note that the interaction of SarA with *agr* results in the promotion of biofilm formation. It was also shown that a mutation of *sarA* resulted in a reduced capacity to form a biofilm, a phenomenon which is independent of the *icaADBC* operon but involves various regulatory pathways, including *sar*, *tcaR*, and sRNA [73,74,75]. Factors that influence staphylococcal biofilm formation have been reviewed by Goetz and Otto [42,76].

The range of environmental factors altering biofilm formation appears to be indicative of the highly diverse habitats in which staphylococci are able to form biofilms. For example, the presence of oleic acid induces *S. aureus* biofilm formation. This probably results from an ionic interaction of the positively charged PIA with the negatively charged oleic acid. The effect is even more pronounced under oxygen-limited conditions [77,78,79], a fact consistent with the observation that anaerobiosis is an important stimulus for *ica* expression [46,80]. A mature biofilm reveals an architecture that ensures the provision of nutrients and oxygen to all cells in the biofilm [3]. As they grow, bacteria begin to arrange in a three-dimensional structure composed of an array of pillars and mushroom-shaped structures. These structures are connected by convoluted channels that deliver nutrients and contribute to the elimination of waste. The maturation of biofilms has been studied by imaging and transcription profiling studies [10,44,81]. A primary discovery that emerged from microarray experiments is that persistence within a mature biofilm requires an adaptive response that limits the deleterious effects of pH reduction associated with anaerobic metabolism [44]. The cell envelope is a very active compartment as the expression of genes that encode binding proteins, proteins involved in the synthesis of murein and glucosaminoglycan, PIA, and other enzymes involved in the cell-envelope metabolism appears to be significantly upregulated. Thus, a biofilm is a dynamic structure that evolves with environmental conditions, such as physical shear forces, and as a result of the processes that are sensed and regulated by the bacteria. Once cell clusters reach a sufficient size, groups of cells either detach (dispersal phase) or die. Thus, it is the cycle of cell growth, detachment, and regrowth that underlies the observed patterns of organized gene expression [51,82].

## 3. Biosynthesis of PIA/dPNAG and Its Regulation

In 1987, Gordon Christensen published a paper on the phenotypic variation of *S. epidermidis* slime production in vitro and in vivo [38]. Today, we know that the “slime” they described was the exopolysaccharide PIA (polysaccharide intercellular adhesin), whose chemical structure was first described in *S. epidermidis* in 1996 [83]. Later, PIA was also referred to as ß-(1,6)-N-acetylglucosamine (PNAG) [84]. The more chemical-sounding name PNAG is not really a correct description of the glucosamine polymer as it ignores the fact that N-deacetylation takes place at certain intervals, which is essential for biofilm formation. PIA represents a linear homoglycan of at least 130 beta-1,6-linked 2-deoxy-2-amino-D-glucopyranosyl residues which are from 80 to 85% N-acetylated. The rest are non-N-acetylated and positively charged. Since a correct chemical description was cumbersome, the name PIA was chosen in the initial description of the structure [83]. PIA is a polymer of partially de-*N*-acetylated ß-1,6-linked *N*-acetylglucosamine (dPNAG). In the scientific community, the terms PIA and PNAG are both used. As this is confusing for the non-specialist reader, we propose to use the term PIA/dPNAG.

## 4. Activity of the *ica* Operon Encoded Enzymes

In the same year that the structure of PIA was published, the corresponding biosynthesis genes, clustered in the *ica* operon, were also identified in *S. epidermidis* [39,40]. With time, it turned out that PIA/PNAG and the corresponding *ica* orthologous genes were not only found in *S. epidermidis* but also in *S. aureus* [60], *Bacillus subtilis* [85], and in many Gram-negative bacteria such as *Acinetobacter baumannii* [86], *Burkholderia* spp. [87], and *Escherichia coli* [88], to name a few. In all these bacteria, PIA/dPNAG contributes to biofilm formation.

The corresponding *icaADBC* orthologous genes were named *epsHIJK* in *B. subtilis* or *pgaABCD* in *E. coli*. The staphylococcal *icaADBC* operon encodes all the enzymes required for the biosynthesis of PIA/dPNAG, as illustrated in Figure 2. The substrate for PIA/dPNAG biosynthesis is UDP-*N*-acetylglucosamine, which is oligomerized by IcaA [89]. IcaA represents the catalytic enzyme that exhibits only a low *N*-acetylglucosaminyltransferase activity, which is possibly enhanced significantly when *icaA* and *icaD* are co-expressed. However, IcaAD reached only a maximal length of 20 residues. When *icaAD* is co-expressed with *icaC*, longer chains are synthesized that react with PIA-specific antiserum. At that time, IcaAD represented a novel protein combination among ß-glycosyl transferases [89].

IcaB is a surface-attached protein that is responsible for the deacetylation of the poly-*N*-acetylglucosamine molecule [90]. Most likely due to the loss of its cationic character, non-deacetylated poly-acetylglucosamine in an isogenic *icaB* mutant strain that is devoid of the capacity to attach to the bacterial cell surface. It is essential for PIA virulence, such as biofilm formation, colonization, and resistance to neutrophil phagocytosis and human antibacterial peptides [90,91].

IcaC is a transmembrane protein containing 18 helices. It is therefore membrane-bound. It demonstrates O-succinyltransferase activity, which is involved in PNAG-O-Succinate addition [92]. This O-succinylation motive constitutes 6% of the succinate molecules of the structure and provides anionic charges to dPNAG as previously described for polysaccharide II, a molecule which demonstrates a lower content in non-N-acetylated D- glucosaminyl residues and contains phosphate and ester-linked succinate, which confer anionic properties [83].

### Regulation of icaADBC Expression

In 1987, Gordon Christensen and colleagues already observed with *S. epidermidis* RP62A (ATCC 35984) that biofilm formation and adherence are not very stable properties [38]. From RP62a, which adheres strongly to glass, they were able to isolate variants with little or no adhesion properties, a phenomenon they termed “phase variation”. The molecular basis of this “phase variation” could be investigated successively after the *ica* operon was found.

## 5. Regulation of icaADBC Expression by Repressors

The *icaR* gene, localized upstream of *icaA* in an inverse orientation, belongs to the *tetR* family of transcriptional regulators. IcaR acts as a repressor of *icaADBC* expression [66]. The target site of IcaR is the 164 nt-long intergenic region (IGR). Within this region, IcaR binds to 42 nt long sequences upstream of the *icaA* start codon. Binding to this region most likely prevents the binding of the RNA polymerase, thus preventing the transcription of *icaADBC* [93]. TcaR, a MarR family of transcriptional regulators of the teicoplanin-associated locus, appears to weakly downregulate the transcription of the *ica* operon, whereas IcaR is a strong negative regulator. Thus, in the absence of *tcaR* and *icaR*, PIA/dPNAG production and biofilm formation was significantly enhanced [74]. Rob, a regulator of biofilm formation, is the third described negative regulator of the *ica* operon [94]. Deletion of *rob* increased the production of PIA/dPNAG. Like IcaR, *Rob* appears to also act as a repressor that binds to the operator site within the *icaR–caA* intergenic region.

## 6. Slipped-Strand Mispairing or “Streisinger Slippage”

The *icaADBC* genes and IGR are rich in repetitive -TATTT- motives. Such tandem motives can lead to frameshift mutations through a mechanism called “Streisinger slippage”, which can result in insertions, deletions, and duplications [95]. One such mutation was first described within the intergenic region, IGR, of the *ica* gene cluster of *S. aureus* SA113 by Jefferson et al. in 2003 [93]. They identified a 5 bp TATTT deletion at the proposed IcaR operator site of IGR. This deletion affected the binding of IcaR and probably other secondary repressors as well, with the effect that *icaADBC* was derepressed. This led to the hyperproduction of PIA/dPNAG and consequently the hyper-mucoid phenotype, as illustrated for MN8m in Figure 3. The 5 bp TATTT deletions with the hyper-mucoid phenotype were also observed in *S. aureus* strains isolated from cystic fibrosis (CF) patients [96]. Mucoid *S. aureus* strains are present in 8.6% of *S. aureus*-positive CF patients, and quite a high proportion of the strains carried a 5bp-deletion (∆-TATTT), suggesting that highly mucoid strains might contribute to the severity of the CF disease [97,98].

The TATT tetranucleotide repeats also play a role in the inactivation of *icaB* and *icaC*. Since PIA/dPNAG-negative *S. aureus* strains are frequently isolated from patients in clinics, a question ensued as to how such variants can arise. Beginning with the hyper-mucoid MN8 strain, PIA/dPNAG-negative variants were isolated and analyzed in more detail. All the mutants JB17, JB15, and JB12 were frameshift mutants in *icaB* and *icaC*, respectively [99]. The JB-mutants suffered either a deletion, such as in JB17 and JB15, or an insertion, such as in JB12, leading each time to a frameshift mutation and the inactivation of the corresponding gene (Figure 3). The slipped-strand mispairing appeared to be reversible and was independent of RecA. The slippage normally occurs during DNA replication and is caused by a DNA polymerase error. This type of deletion event of nucleotides is common in many organisms and can be advantageous when it activates beneficial genes that enhance microbial survival in adverse environments; however, it can also be deleterious when it alters or suppress function of genes relevant to survival [100].

Another *ica* regulatory mechanism was described in *S. epidermidis*. One of the first findings was an alternating insertion and excision of the insertion sequence element IS256 [101]. More recently, a long, non-coding (nc) RNA named *icaZ* was found to exclusively exist in *ica*-positive *S. epidermidis* strains such as O47 or RP62A, but not in *S. aureus* or other staphylococci [102]. *icaZ* blocks *icaR* mRNA translation, causing the derepression of the *icaADBC* operon, thus causing increased PIA/dPNAG production (Figure 3). For *S. epidermidis*, PIA/dPNAG appears to play an even greater role in colonization and biofilm-associated infections than for *S. aureus*, which has an additional arsenal of surface proteins that support colonization and biofilm formation.

In *S. aureus* and most likely other staphylococcal species, the overexpression of PIA/dPNAG appears to be beneficial only under certain infectious conditions, such as in biofilm-associated infections in which it is crucial that the pathogen adheres to the tissue or implant material, or in CF in the lung, where PIA/dPNAG can protect bacteria from phagocytosis and certain antibiotics. *S. aureus* is also a leading cause of prosthetic joint infections (PJI) in which biofilm-associated organisms demonstrated recalcitrance to immune-mediated clearance and antibiotic susceptibility [103]. The authors of the latter study discovered no polymorphisms in the IGR or *icaBC* genes but found an SNP within the *icaR* coding region. This resulted in a V176E change in IcaR that affected its binding activity, resulting in increased *icaADBC* operon transcription and PIA/dPNAG production.

During infection, bacteria are exposed to different stress situations. Thus, a hyper-biofilm former may have an advantage in implant-associated infections but probably not in sepsis. However, since the different types of infection usually occur simultaneously or sequentially during an infection, we usually isolate a mixture of *S. aureus* strains that range from non-biofilm to hyper-biofilm strains when taking samples in the clinic. Most likely, a moderate mucoid state is the normal state in *S. aureus* and *S. epidermidis.* However, a hyper-mucoid phenotype is advantageous only under specific infectious conditions, particularly colonization and survival. Perhaps this is why so many mechanisms for controlling and fine-tuning *ica* expression have evolved. However, under normal growth conditions, the hyperproduction of PIA/dPNAG causes a fitness burden, and PIA/dPNAG-negative strains have a growth advantage and overgrow PIA/dPNAG-positive strains [99]. 

## 7. Roles of Biofilm in the Tolerance to Multiple Drugs

In a biofilm, the bacterial cells are attached to a surface where, depending on the nutrient content of the environment, they multiply more or less actively and form a multilayered structure. The maturation to a three-dimensional biofilm is also called the accumulation phase. Such biofilms are formed in humid or marine environments in water pipes, on ship hulls, and other on stainless steel surfaces where they cause biofouling [104], which causes enormous costs [105,106]. Typically, such a biofilm consists of a heterogeneous spectrum of micro- and macro-organisms whose cells are embedded in a self-produced matrix and whose metabolic products lead to the corrosion of the metal [107]. In particular, the production of extracellular polymeric substances (EPSs) by microorganisms facilitates adhesion to material surfaces such as metals. These complex biofilm structures are highly resistant to extreme stress conditions, and only aggressive bactericidal detergents or harsh physical treatments such as sonication exhibit antifouling properties [108].

There are similarities and differences between biofouling and biofilm-associated infections. They have in common that microorganisms primarily bind to surfaces and change these surfaces by their binding so that further microorganisms can bind and thus form a robust biofilm, whereby EPSs make an important contribution to the compactness of the biofilm. While biofouling is a mixture of various microorganisms, biofilm-associated infection is usually due to a single bacterial species. The National Institutes of Health (NIH) evaluated that biofilm-producing bacteria are involved in 65% of all microbial infections and are responsible for 80% of chronic infections. The annual incidence of biofilm-related infections in the United States represents roughly 2 million cases, causing 268,000 estimated deaths, and is accompanied by USD 18 billion in direct costs for the therapy of these infections [2,109]. The bacterial species frequently involved in such infections are *S. epidermidis*, *S. aureus*, *Enterococcus*, *Bacillus*, and *Candida* spp. The origin of these microorganisms may be from the skin or from other indwelling devices such as central venous catheters or dental work [110]. 

With biofilm-associated infection, the largest problem is that many therapeutic approaches fail because a high proportion of the bacterial cells in a biofilm matrix are “phenotypically” insensitive to most antibiotics. We deliberately speak here not of resistance, since the latter implies certain resistance genes in the classical sense. In 1994, after penicillin was marketed, it was observed that staphylococci can enter a physiological state called persistence (or multidrug tolerance) in which lethal antibiotics failed to kill them [111]. Multiple factors appear to contribute to the global insensitivity of biofilm bacteria [13,112]: Enhanced antimicrobial resistance is a general phenomenon of biofilms and is the result of numerous specific factors which depend on the species involved, the environment of the biofilm, and the antimicrobial agent used;The implant material on which a biofilm is formed is not or is only scarcely perfused, preventing antibiotic diffusion at a sufficiently high concentration;The penetration and diffusion of antibiotics into a thick biofilm is hampered;The growth rate of bacterial cells in a biofilm is reduced (most antibiotics are efficient against actively growing bacteria);The physiology of cells in a biofilm differs from that of planktonic cells.

The phenomenon of the general antibiotic insensitivity of bacterial cells in a biofilm is characterized by the fact that biofilm-associated cells are insensitive, whereas “the same” cells in suspension are sensitive [113]. This suggests that insensitivity is not related to classical antibiotic resistance gene but to an altered physiological state in the biofilm mode of growth. Kim Lewis called the small fraction of essentially invulnerable cells in a biofilm “persisters” that exhibit multidrug tolerance (MDT) [114]. In *Escherichia coli*, the toxin–antitoxin (TA) modules RelE-RelB and HipB-HipA (high-persistence) seam to play a role in the persister phenotype. The overproduction of RelE or HipA causes an increase in the persister population. HipA inhibits translation by the phosphorylation of EF-Tu [115], stimulates the RelA-dependent synthesis of (p)ppGpp [116], and phosphorylates glutamyl-tRNA synthetase (GltX), which becomes inactivated by phosphorylation by HipA [117]. RelE cleaves mRNA at the ribosomal A site with high codon specificity [118]. The overexpression of RelE or HipA leads to a slowdown translation and thus the growth of *E. coli*, which presumably protects the cells from lethal factors such as antibiotics. It is known from ß-lactam antibiotics that they act mainly on dividing cells and are less effective on non-growing cells.

In staphylococci, the generation of persister cells is less clear than in *E. coli*. There are four different families of TA systems described, but their physiological roles are elusive [119]. The chromosomal *mazEF* system encodes the RNase toxin MazF and the antitoxin MazE [120]. MazF specifically targets UACAU sequences of *spa* (staphylococcal protein A) and *rsbW* (anti-sigmaB factor) in *S. aureus* mRNA in vivo, whereas translational reporter fusions indicated that the protein levels of the encoded products were unaffected. Despite a comparable growth rate to the wild-type, an *S. aureus mazEF* deletion mutant was more susceptible to β-lactam antibiotics, suggesting that the genes involved in antibiotic stress response or cell wall metabolism are controlled by this TA system [120].

Long before *E. coli*, a connection between reduced growth and increased antibiotic tolerance was described in staphylococci in the form of “small colony variants” (SCVs) [121]. From patients with persistent and relapsing infections, *S. aureus* SCVs were isolated which were auxotrophs for menadione, hemin, and/or a CO_2_ supplementation. All these SCVs were resistant to aminoglycosides. The phenotype of such respiratory deficient mutants was further analyzed in a stable *hemB* mutant of *S. aureus* [122]. Such a *hemB* mutant showed the typical SCV phenotype, such as slow growth and a resistance to aminoglycosides; it also showed decreased pigmentation, low coagulase activity, reduced hemolytic activity, and a high persistence in endothelial cells. Respiratory mutants, both those that are naturally occurring or genetically constructed, demonstrate the importance of the metabolism in virulence and drug tolerance [123]. In *S. aureus*, there are many global regulators that impact virulence factor expression in SCVs [124].

## 8. A Glimpse into Staphylococcal Biofilm Matrix and Physiology

Durable catheters are most frequently colonized with staphylococci, followed by *Acinetobacter calcoaceticus* or *Pseudomonas aeruginosa*. The first microscopic studies of a staphylococcal biofilm were performed by the group of Georg Peters [125]. Scanning electron microscopy (SEM, magnification × 5000) of *S. epidermidis* grown for 24 h on a cellulose acetate surface showed closely packed bacterial cells embedded in a slimy matrix. An image of such an *S. epidermidis* biofilm on a catheter is shown in Figure 4A. We now know that the slimy matrix consists of the exopolysaccharide PIA/dPNAG. The question that concerned the researchers was which metabolism would be predominant in such a biofilm. It was shown that an *S. aureus sarA* (staphylococcal accessory regulator A protein) mutant was impaired in biofilm formation. A transcriptome analysis with this mutant suggested that cells grow essentially anaerobically in mature biofilms cells, and that the genes of the acid tolerance response, such as the global regulators SigB and SarA, are upregulated in response to an acidic (pH 5.5) environment [44].

A comparative transcriptome analysis between biofilm- and planktonic-grown *S. aureus* showed that the envelope appeared to be a very active compartment in biofilm-associated cells. Indeed, genes that encode binding proteins, proteins involved in murein and PIA/dPNAG synthesis, and enzymes involved in cell envelope metabolism were significantly upregulated [81]. In addition, formate fermentation (formate dehydrogenase), urease activity, the response to oxidative stress (staphyloxanthin), acid and ammonium production, and the arginine deaminase cluster were upregulated in a biofilm. On the other hand, toxins and proteases were upregulated under planktonic growth conditions. Interesting, the *ica* operon was highly upregulated during the first 8 h of biofilm growth compared to planktonic organisms. The expression level in the biofilm cells then decreased, but remained 3-fold higher than the expression level in a planktonic state. It has been assumed that enzymes have a long half-life [126]; therefore, the upregulation of these genes might not be needed in aged, surface-associated cells, as biofilm formation has begun and cell growth is retarded due to nutrient depletion [81]. A comparative proteome analysis essentially confirmed the transcriptome results [10]. Compared to planktonic growth, biofilm cells expressed higher levels of proteins associated with cell adhesion, peptidoglycan synthesis, fibrinogen-binding proteins, and enzymes involved in pyruvate and formate metabolism as well as SarA, which is in accordance with the positive effect of SarA on *ica* locus expression.

All data indicate that anoxic conditions prevail in the biofilm, as also indicated by the upregulation of pyruvate formate lyase (Pfl) and NAD-dependent formate dehydrogenase (Fdh) in *S. aureus* biofilms. To investigate their physiological role, *fdh* and *pfl* deletion mutants were constructed (Δ*fdh* and Δ*pfl*) and their impact on the biofilm was analyzed [127]. The absence of formate production was recorded in the *pfl* mutant, and glucose consumption was delayed. Thus, as ethanol production was decreased, acetate and lactate production were unaffected. All metabolic alterations could be restored by the addition of formate or complementation of the Δ*pfl* mutant. All results suggest the model proposed in Figure 4B. The upregulation of *pfl* takes place in the deeper layer of the biofilm where anoxic conditions are prevalent (Pfl is oxygen sensitive) and nutrients are limited. Pfl is necessary under these conditions, allowing for the formation of C_1_ units (formate) for formyl-THF synthesis and for protein and purine biosynthesis. Interestingly, not only *fdh* and *pfl* but also the formyl-THF synthetase gene (*fhs*) were upregulated under biofilm conditions [81]. It was suggested that Fdh plays a role in the microaerobic area of the biofilm. Formate is produced in the anaerobic area by Pfl and diffuses to the microaerobic region, where it is oxidized by Fdh to produce CO_2_ and NADH/H^+^. Formate is then detoxified, and NADH/H^+^ can be respired in the presence of small amounts of oxygen and no longer comprises a burden for *S. aureus*. The importance of Pfl under anoxic conditions lies in its ability to supply the cells with sufficient formate, which is used via formyl-THF for protein and purine synthesis. The consequence is that in the *pfl* mutant, the amount of fMet-polypeptides produced was lower than in the strain. Based on these findings and on the benefits of functional *pfl*, *fdh*, and *fhs*, we concluded that the upregulation of these genes might represent an important survival strategy in the biofilm mode of growth [127].

**Figure 4 ijms-24-05218-f004:**
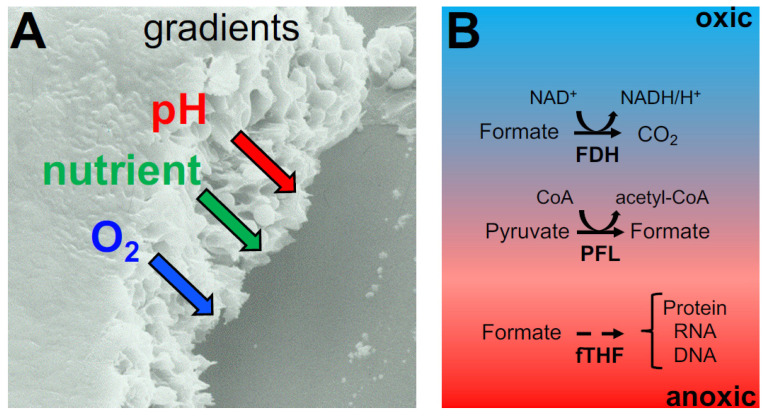
Biofilm matrix and metabolism. (**A**) Scanning electron microscopy (SEM), of *S. epidermidis*. The cells grown for 24 h on a cellulose acetate surface show closely packed bacterial cells embedded in a slimy matrix [125]. The cells are embedded in many layers in this biofilm. The bacterial cells multiply upwards from the adherent cells and are surrounded by a layer of mucus to create a multilayer coating (biofilm). The picture shows a cross-section of this biofilm. At the perfused surface, the pH is neutral and nutrients and oxygen are present. However, towards the layers below, the oxygen, nutrients, and pH decrease continuously, forming gradients. It can be assumed that the cells permanently adapt to the changed conditions and released metabolites. (**B**) Schematic representation of formate metabolism in *S. aureus* biofilms. In the anaerobic layers (red) of the mature biofilm, the PFL converts pyruvate to acetyl-CoA and formate. The latter can be used by strictly anaerobically grown cells for the synthesis of formyl-THF and therefore for the biosynthesis of proteins, DNA, and RNA. At the same time, formate accumulates and diffuses to microaerobic regions (light red). Here, it might be oxidized by the FDH under the production of NADH [127].

## 9. Staphylococcal Biofilm in the Clinical Situation

At the end of the 1990s in the United States, experts estimated that biofilms were associated with 65% of nosocomial infections and that the annual cost of treatment of these biofilm-associated infections was higher than USD 1 billion [2,128]. *S. aureus* and other staphylococci are frequently found on implanted materials such as catheters, hip prosthesis, or surgical materials [5,25,129,130]. A recent study identified methicillin-resistant coagulase negative staphylococci as a major cause of biofilm-associated infections and possibly responsible for critical clinical situations. This interesting study relied on the analysis of numerous samples originating from hospital environments and from various hospital wards. The authors identified different staphylococcal species that produce bacterial biofilms: *Staphylococcus haemolyticus*, *S. epidermidis*, *S. hominis*, and *S. warneri*. The authors isolated approximately 300 MR-CoNS among the 558 samples from community and hospital environments. *S. haemolyticus* and *S. epidermidis* were the predominant species, representing roughly 73% of the CoNS identified. Significant biofilm production was detected in 91% of isolates, suggesting that the absence of production is marginal in clinical and environmental CoNS [131]. The staphylococci isolates that were derived from hospital wards were more associated with biofilm production than the community-derived isolates. Distinguished from the isolates identified in hospital wards, environmental strains were devoid of *icaAD* and *bap* genes and thus produced mainly proteinaceous biofilms. 

Recent studies documented biofilms as community phenomena by assessing the interaction between bacteria and surface-associated-biofilm-producing organisms. Toledo-Silva reported nicely that numerous non-*aureus* species of staphylococci were able to interact with biofilm-producing *S. aureus*. The authors isolated *S. chromogenes*, *S. epidermidis*, and *S. simulans* from bovine milk samples and showed that *S. chromogenes* (devoid of *ica*) stimulates the biofilm formation of *S. aureus* and alters the dispersion of *S. aureus*-formed biofilm. The study highlighted possible interactions between CoNS and *S. aureus* in the biofilm communities, most likely through interactions between the respective *agr* quorum systems [132]. Further research is needed to study bacterial biofilms as community phenomena.

## 10. Recent Attempts to Reduce/Destroy a Biofilm during Infection

Dozens of recent papers described different attempts to reduce biofilm formation or to destroy an already-formed biofilm in order to avoid the removal or replacement of an implanted material. Enzymes, antimicrobial peptides, bacteriophages, and natural compounds from vegetal origins have been used, mainly in vitro, with variable results [133,134,135,136,137,138,139,140,141,142,143,144,145]. Interesting observations were reported by Caballero Gomez in the field of meat-chain production using natural compounds extracted from essential oils alone or in combination with EDTA [133]. The authors described a significant effect in terms of the antimicrobial capacity of thymol, limonene, geraniol, or eugenol and an important inhibition of the biofilm formation of *S. aureus*, *Enterococcus*, and *Pseudomonas* when these molecules were used in combination with EDTA. A similar strategy of potentiation of biocides by EDTA was used successfully on an already-formed biofilm [140]. These strategies appeared to be more adapted to the treatment of surfaces or devices than to medical materials. Other natural molecules have been identified from human milk, such as oligosaccharides, that allow for an appreciable decrease in *S. aureus* biofilm formation [136]. Other groups reported the efficacy of different synthetic molecules of an organic [134,139] or peptidic [135] nature on biofilm formation with clear success. Impressive results were obtained using a non-toxic acyclic amine derivative that yielded an extensive reduction of a biofilm and bacterial count in a model of urinary catheter infection [139]. The hypotheses of these studies relied on the alteration of the regulation of the biosynthesis process of bacterial biofilm, especially through the inhibition of the quorum-sensing system, which is an appealing solution. 

Based on the observation that antibiotics alone have almost no effect on the destruction of already-formed biofilms, Liu and colleagues combined various molecules to reduce the amount of biofilm and embedded bacteria on a material. The most impressive effect was observed with combinations of oxytetracycline and subtilisin A or oxytetracycline and calcium gluconate [146]. Divalent cations, such as Ca^2+^,and proteins possibly have a significant role in the structuration of the extracellular matrix that constitutes a bacterial biofilm, and the combination of these compounds with oxytetracyline resulted in a synergistic effect of killing and detachment [146]. A similar synergism was observed when fusidic acid and the quaternary ammonium berberine chloride were used, even on fusidic-acid-resistant MRSA isolates [147]. An interesting study reported various effects of baicalein—an inhibitor of the cytochrome p450 system—on the alteration of toxin expression in *S. aureus* and on biofilm formation [148]. The utilization of baicalein and vancomycin yielded a drastic reduction in biofilm formation and bacterial viability. The authors suggested a baicalein-mediated mechanism that yields the disruption of mature biofilms and increases the permeability of the bacterial envelope to antibiotics [148].

Bacterial biofilms are composed mainly of polymeric molecules: DNA, polysaccharides, and proteins. Hydrolytic enzymes are also potentially interesting molecules. Gutierrez and colleagues obtained very interesting results by using an engineered endolysin treatment in vitro and in vivo in a skin model of infection. The authors obtained a complete disinfection of the contaminated sites after treatment [138]. Similar results were envisioned by using lytic bacteriophages on already-formed biofilms. A recent study by Pallavali and colleagues illustrated nicely the potential of active phage particles. The authors obtained impressive results following a single treatment of already-formed biofilms for 24–96 h. These different attempts at biofilm eradication support the possible development of alternative strategies to antibiotics for the treatment of contaminated biomaterials used in a clinical context.

## 11. Concluding Remarks

The majority of bacterial pathogens involved in nosocomial infections are able to produce a biofilm. *S. aureus*, *S. epidermidis*, and other staphylococci are frequently found on implanted material such as catheters and hip prostheses. The different factors involved in the regulation of biofilm production as well as in biofilm biology represent particularly active fields of research that allow for the development of potentially usable strategies to eradicate biofilms in a clinical setting in order to avoid the removal of colonized biomaterials.

## Figures and Tables

**Figure 1 ijms-24-05218-f001:**
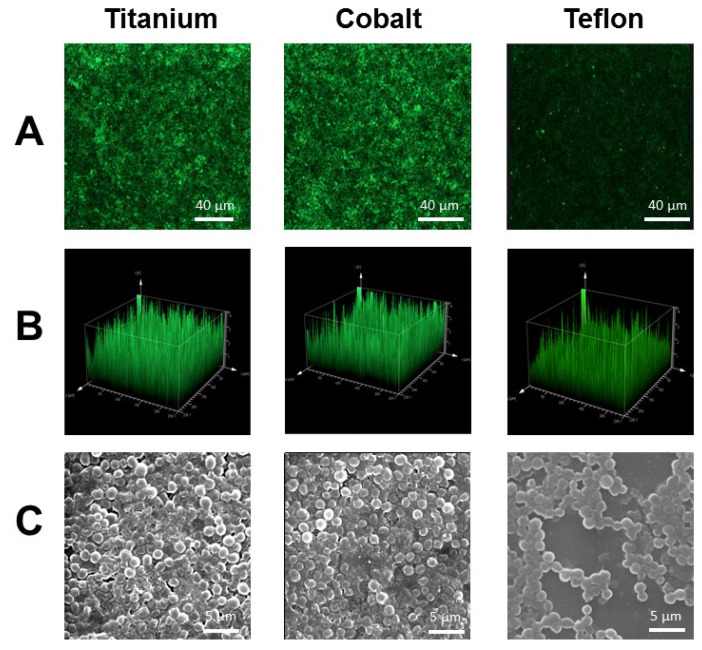
Biofilm formation of *S. aureus* (pCtuf-gfp) on glass slides coated with different biomaterials. (**A**) Confocal scanning laser micrographs of *S. aureus* (pCtuf-gfp) biofilm formation on glass slides coated with titanium, cobalt–chromium, and amorphous Teflon. (**B**) Three-dimensional view of the fluorescence emitted by *S. aureus* (pCtuf-gfp) in the biofilm. (**C**) Scanning electron micrographs (SEM) of the corresponding biofilms. *S. aureus* adheres very strongly to surfaces coated with titanium and cobalt–chromium, yielding thick biofilms, while adherence to Teflon was decreased and a less-dense biofilm was formed (modified according [21]).

**Figure 2 ijms-24-05218-f002:**
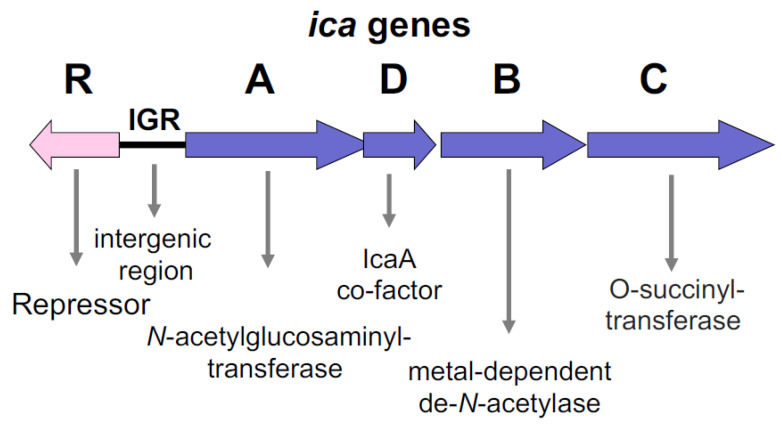
Organization and function of the *ica* gene cluster in staphylococci. The cluster is composed of the *ica* operon *icaADBC* and the repressor gene *icaR*, which is inversely oriented to *icaA*. The approximately 160–170 nt long intergenic region (IGR) carries the promoters for *icaR* and *icaA* and operator sites. The *icaADBC* operon encodes all enzymes necessary for PIA/dPNAG biosynthesis. IcaA is a cytoplasmic enzyme which has *N*-acetylglucosaminyltransferase activity using UDP-*N*-acetylglucosamine as a substrate; its activity is enhanced by IcaD, which acts as a co-enzyme. IcaB is a surface-attached poly-*N*-acetylglucosamine deacetylase responsible for deacetylation of approximately every fourth *N*-acetylglucosamine molecule; its activity is essential for biofilm function. IcaC is membrane-localized and demonstrates O-succinyltransferase activity in approximately 6% of the dPNAG, rendering PIA/dPNAG anionic. IcaC plays also a role in the elongation of oligo *N*-acetylglucosamines.

**Figure 3 ijms-24-05218-f003:**
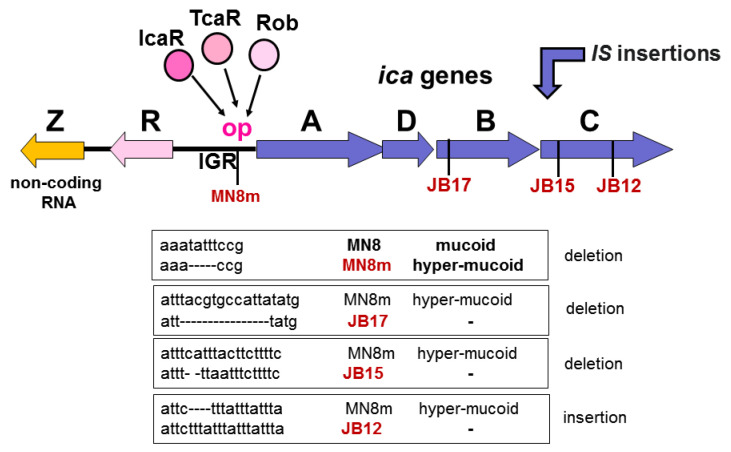
Phase variation of staphylococcal PIA/dPNAG expression. *icaADBC* expression is controlled at different levels. (1) Repressor binding: IcaR binds to the operator site (op) of the intergenic region (IGR) and thus prevents *icaADBC* transcription by blocking RNA polymerase binding. TcaR and Rob also appear to bind to IGR, but the exact binding needs to be verified. (2) Repetitive -TATT- motives cause deletions and insertions by slipped-strand mispairing during DNA replication. A 5 bp TATTT deletion in IGR affects binding of the repressor proteins (IcaR, TcaR, and Rob) causing overexpression of *icaADBC*. The same motive in *icaB* and *icaC* causes frameshift mutations by small deletions or insertions, leading to gene inactivation. (3) In *S. epidermidis*, there is downstream of the *icaR* gene, *lacZ*, that encodes a non-coding RNA which silences icaR expression, causing *icaADBC* activation. (4) Finally, insertion sequences (IS) can integrate in *ica* genes, causing inactivation of the corresponding gene.

**Table 1 ijms-24-05218-t001:** Staphylococci can colonize nearly any material.

Materials of Prosthetic Devices	References
Poly-ethylenetetraphtalate, poly-propylene amorphous. silicone rubber, poly-tetrafluoroethylene,	[22]
poly-propylene crystalline,	
poly-vinylidine fluoride, polyesther urethane, polyethylene, cellulose acetate, polycarbonate,	
polyester urethane	
Pacemaker lead	[23]
Cyanoacrylate (n-butyl-2-cyanoacrylate)	[24]
Tissue adhesive/Robbins device	
Surgical-grade biomaterials:	[25]
stainless steel, aluminum ceramic, methyl methacrylate and high-density polyethylene	
Intravascular catheters:	[26]
thermoplastic polyurethane, silicone elastomer	
polyurethane coated with hydromer, serum coating of catheters	
Breast prostheses:	[27,28,29]
silicone; silicone and polyurethane foam; silicone breast implant	
Teflon catheter	[30]
Poly(methyl methacrylate)	[31]
Silk threads	[32]
Contact lenses:	[33]
polymacon, etafilcon A, vifilcon A	
Stainless steel, orthopedic nuts	[34]
Dentures	[35]
Polystyrene	[36]
Titanium:	[37]
stainless steel; cortical bone surfaces	
Glass	[38,39,40]

## Data Availability

Data sharing not applicable. No new data were created or analyzed in this study. Data sharing is not applicable to this article.

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
