# Peer review of "Biology and Regulation of Staphylococcal Biofilm"

_ijms, 2023, doi:10.3390/ijms24065218_

Round 1
Reviewer 1 Report
François et. al., described the role of the different factors involved in the regulation of biofilm formation which might help researchers to develop new strategies to eradicate biofilms. I have few suggestions if authors can address to strengthen their work,
1. As we can see many drugs are not effective against biofilm inhibition and becoming resistant and thereby, I suggest incorporating alternate strategies like synergistic study in biofilm inhibition in the recent attempts to reduce/destroy biofilm during infection part.
2. I suggest representing a schematic diagram of the review article that might help readers.
Author Response
Response to Reviewer 1 Comments
- As we can see many drugs are not effective against biofilm inhibition and becoming resistant and thereby, I suggest incorporating alternate strategies like synergistic study in biofilm inhibition in the recent attempts to reduce/destroy biofilm during infection part.
AU1: An additional paragraph has been introduced in the revised version of the article as well as some new references.
- I suggest representing a schematic diagram of the review article that might help readers.
AU2: I am not convinced to understand the point as the document already contains 6 figures including 2 schemes describing interactions of the whole ica locus.
Reviewer 2 Report
The submitted article “Biology and regulation of Staphylococcal biofilm” by Patrice François, Jacques Schrenzel and Friedrich Götz, is a review article focused on composition and regulation of biofilm development in staphylococci. Moreover, the authors say they summarize recent studies reporting various attempts to destroy already formed biofilm.
The topic has clinical importance, because biofilm-producing pathogens are responsible for many human diseases which are difficult to treat, especially those involving infection of biomaterials such as catheters and other implants. The authors are recognized researchers with lifelong experience in the field of bacterial biofilm regulation. Accordingly, the topic is clearly presented and discussed with the support of appropriate references, in the following sections:
“Biofilm formation and clinical significance” (Line 27); “Molecular Control of S. aureus Biofilm Development and the role of ica” (line 93); “Biosynthesis of PIA/dPNA and its regulation” (line 164); “Activity of the ica operon encoded enzymes” (line 179); “Regulation of icaADBC expression” (line 221); “Regulation of icaADBC expression by repressors.” (line 228); “Slipped-strand mispairing or 'Streisinger slippage'.“ (line 242); “Roles of biofilm in the tolerance to multiple drugs” (line 313).
Regarding these sections, I have only minor comments, which are listed below.
Major concern
On the contrary, I have a major concern regarding the section “A glimpse into staphylococcal biofilm matrix and physiology” (line 394). The text between lines 394 and 449 is clearly presented, including Figure 4, and discussed according to the cited references. However, it is not clear whether data presented after Figure 4, i.e. in figure 5 and figure 6, and in the accompanying text (lines 466-574), are already published or not. In the first case, the respective reference(s) should be provided. In the second case, in my opinion, presenting unpublished data in a review article is not correct, as in this way the authors avoid peer reviewing of their experimental work. In my opinion, the authors should clearly state that data they discuss are unpublished. In addition, as the amount of these data is considerable (growth kinetics, qPCR analysis, fluorescence microscopy, transcriptomic profile), it would be very useful for the reviewer if the authors provide more experimental details in a separate file.
I would like to know the Editors' opinion about this.
Minor comments
The section “Bacterial biofilm in the clinical situation” (line 575) should be improved. For instance, the statement on line 576 is a repetition of the statement on lines 87-88 and can be removed.
Lines 584, 586, 588: abbreviations should be uniform (CoNS and CNS both refer to coagulase-negative staphylococci?) and provide a list of abbreviations;
The style of the section titles should be uniform (e.g. or all titles in bold, or all titles in italics, or all titles numbered, etc, according to the journal’s style and guidelines).
Line 164: correct “PIA/dPNAG” (G was missing);
Line 193: delete “amino acid”;
Line 197: replace “moieties” with “protein”;
lines 205-206: the following statement should be corrected as something is not clear from a syntactic point of view: “This O-succinylation modification constitutes 6% succinate molecule provides anionic charges to dPNAG…”;
Line 208: delete “a” in the construction “conferring anionic properties”;
Line 234: add “of” in the construction “a MarR family of transcriptional regulators of the teicoplanin- associated locus, …”;
Line 245: correct “mutation” (singular);
Line 250: correct “illustrated”;
Figure 3: apart from “Rob” and “IcaR”, “TcaR” should be added?;
line 283: singular;
line 284: plural;
line 298: replace “they” with “the authors of the latter study”;
line 300: delete the second “in”;
line303: correct “advantage” instead of “advantageous”;
lines 332-333: in the construct “biofilm-producing bacteria are involved in 65% and 80% of all microbial and chronic infections, respectively”, it is not clear to me what is the difference between “microbial” and “chronic”. Please, check.
Line 360: singular;
line 385: add “supplement” after “CO2”;
line 468: a reference is missing;
lines 469-470: add the reference after the construct “as previously described”;
The list is not exhaustive, there are other errors appearing now and then, so, please, read the text carefully and check for errors (english and typos).
Line 796: delete reference #70 which is a duplicate of reference #43.
Author Response
Response to Reviewer 2 Comments
Minor comments:
The section “Bacterial biofilm in the clinical situation” (line 575) should be improved. For instance, the statement on line 576 is a repetition of the statement on lines 87-88 and can be removed.
AU: Repetition has been removed accordingly. The paragraph has been modified following reviewer recommendation and the title of the paragraph slightly modified as well.
Lines 584, 586, 588: abbreviations should be uniform (CoNS and CNS both refer to coagulase-negative staphylococci?) and provide a list of abbreviations;
AU: CoNS is now used in the document. A list of all abbreviations used in the review has been added at the end of the document.
The style of the section titles should be uniform (e.g. or all titles in bold, or all titles in italics, or all titles numbered, etc, according to the journal’s style and guidelines).
AU: modified accordingly
Line 164: correct “PIA/dPNAG” (G was missing);
AU: modified accordingly
Line 193: delete “amino acid”;
AU: modified accordingly
Line 197: replace “moieties” with “protein”;
AU: modified accordingly
lines 205-206: the following statement should be corrected as something is not clear from a syntactic point of view: “This O-succinylation modification constitutes 6% succinate molecule provides anionic charges to dPNAG…”;
AU: modified in the revised version of the MS
Line 208: delete “a” in the construction “conferring anionic properties”;
AU: modified accordingly
Line 234: add “of” in the construction “a MarR family of transcriptional regulators of the teicoplanin- associated locus, …”;
AU: modified as suggested
Line 245: correct “mutation” (singular);
AU: modified as suggested
Line 250: correct “illustrated”;
AU: modified as suggested
Figure 3: apart from “Rob” and “IcaR”, “TcaR” should be added?;
AU: Added as suggested
line 283: singular;
AU: modified as suggested
line 284: plural;
AU: modified as suggested
line 298: replace “they” with “the authors of the latter study”;
AU: modified as suggested
line 300: delete the second “in”;
AU: modified as suggested
line303: correct “advantage” instead of “advantageous”;
AU: modified as suggested
lines 332-333: in the construct “biofilm-producing bacteria are involved in 65% and 80% of all microbial and chronic infections, respectively”, it is not clear to me what is the difference between “microbial” and “chronic”. Please, check.
AU: modified accordingly
Line 360: singular;
AU: modified as suggested
line 385: add “supplement” after “CO2”;
AU: modified as suggested
line 468: a reference is missing;
AU: reference added
lines 469-470: add the reference after the construct “as previously described”;
AU: reference has been added
The list is not exhaustive, there are other errors appearing now and then, so, please, read the text carefully and check for errors (english and typos).
Line 796: delete reference #70 which is a duplicate of reference #43.
AU: reference has been deleted
Major concern
On the contrary, I have a major concern regarding the section “A glimpse into staphylococcal biofilm matrix and physiology” (line 394). The text between lines 394 and 449 is clearly presented, including Figure 4, and discussed according to the cited references. However, it is not clear whether data presented after Figure 4, i.e. in figure 5 and figure 6, and in the accompanying text (lines 466-574), are already published or not. In the first case, the respective reference(s) should be provided. In the second case, in my opinion, presenting unpublished data in a review article is not correct, as in this way the authors avoid peer reviewing of their experimental work. In my opinion, the authors should clearly state that data they discuss are unpublished. In addition, as the amount of these data is considerable (growth kinetics, qPCR analysis, fluorescence microscopy, transcriptomic profile), it would be very useful for the reviewer if the authors provide more experimental details in a separate file.
I would like to know the Editors' opinion about this.
AU: Authors would like to give some precisions about this observation. Submission of original data in this type of special issue is not rare and is definitely not a way to avoid reviewing, as the article is reviewed. The microarrays used in this report have been extensively described and validated in the past, for genome content analysis and transcriptomic studies [1-10]. Reference cited allows readers to obtain the experimental section used for that purpose [1]. We clearly indicate in the revised version of the paper that the results have not been published and reflect only our observations, which are consistent with data presented with this pair of strains in previous papers published by the group of the last author of the present review article. We are totally confident with editor position and will respect her choice.
- Charbonnier, Y.; Gettler, B.M.; Francois, P.; Bento, M.; Renzoni, A.; Vaudaux, P.; Schlegel, W.; Schrenzel, J. A generic approach for the design of whole-genome oligoarrays, validated for genomotyping, deletion mapping and gene expression analysis on Staphylococcus aureus BMC Genomics 2005, 6, 95.
- Reichert, S.; Ebner, P.; Bonetti, E.J.; Luqman, A.; Nega, M.; Schrenzel, J.; Sproer, C.; Bunk, B.; Overmann, J.; Sass, P.; et al. Genetic Adaptation of a Mevalonate Pathway Deficient Mutant in Staphylococcus aureus. Front Microbiol 2018, 9, 1539, doi:10.3389/fmicb.2018.01539.
- Jonsson, I.M.; Juuti, J.T.; Francois, P.; AlMajidi, R.; Pietiainen, M.; Girard, M.; Lindholm, C.; Saller, M.J.; Driessen, A.J.; Kuusela, P.; et al. Inactivation of the Ecs ABC transporter of Staphylococcus aureus attenuates virulence by altering composition and function of bacterial wall. PLoS.ONE. 2010, 5, e14209.
- Koessler, T.; Francois, P.; Charbonnier, Y.; Huyghe, A.; Bento, M.; Dharan, S.; Renzi, G.; Lew, D.; Harbarth, S.; Pittet, D.; et al. Use of Oligoarrays for Characterization of Community-Onset Methicillin-Resistant Staphylococcus aureus Journal of Clinical Microbiology 2006, 44, 1040-1048.
- Mee-Marquet, N.; Corvaglia, A.R.; Valentin, A.S.; Hernandez, D.; Bertrand, X.; Girard, M.; Kluytmans, J.; Donnio, P.Y.; Quentin, R.; Francois, P. Analysis of prophages harbored by the human-adapted subpopulation of Staphylococcus aureus CC398. Infect.Genet.Evol. 2013, 18, 299-308.
- Pietiainen, M.; Francois, P.; Hyyrylainen, H.L.; Tangomo, M.; Sass, V.; Sahl, H.G.; Schrenzel, J.; Kontinen, V.P. Transcriptome analysis of the responses of Staphylococcus aureus to antimicrobial peptides and characterization of the roles of vraDE and vraSR in antimicrobial resistance. BMC.Genomics. 2009, 10, 429.
- Pohl, K.; Francois, P.; Stenz, L.; Schlink, F.; Geiger, T.; Herbert, S.; Goerke, C.; Schrenzel, J.; Wolz, C. CodY in Staphylococcus aureus : a regulatory link between metabolism and virulence gene expression. J Bacteriol. 2009, 191, 2953-2963.
- Pritchard, L.; Liu, H.; Booth, C.; Douglas, E.; Francois, P.; Schrenzel, J.; Hedley, P.E.; Birch, P.R.; Toth, I.K. Microarray comparative genomic hybridisation analysis incorporating genomic organisation, and application to enterobacterial plant pathogens. PLoS.Comput.Biol. 2009, 5, e1000473.
- Renesto, P.; Rovery, C.; Schrenzel, J.; Leroy, Q.; Huyghe, A.; Li, W.; Lepidi, H.; Francois, P.; Raoult, D. Rickettsia conorii transcriptional response within inoculation eschar. PLoS.ONE. 2008, 3, e3681.
- Resch, G.; Francois, P.; Morisset, D.; Stojanov, M.; Bonetti, E.J.; Schrenzel, J.; Sakwinska, O.; Moreillon, P. Human-to-bovine jump of Staphylococcus aureus CC8 is associated with the loss of a beta-hemolysin converting prophage and the acquisition of a new staphylococcal cassette chromosome. PLoS One 2013, 8, e58187, doi:10.1371/journal.pone.0058187
Round 2
Reviewer 2 Report
The revised version of the submitted review article is now suitable for publication in IJMS. Before publication, please check the manuscript carefully for small errors and typos.
I appreciate the Editors' and authors' agreement on the decision to remove the unpublished data from this review article.